# Self-reported and measured anthropometric variables in association with cardiometabolic markers: A Danish cohort study

Jie Zhang[1]*, Anja Olsen[1,2], Jytte Halkjær[2], Kristina E. Petersen[2], Anne Tjønneland[2], Kim Overvad[1], Christina C. Dahm[1]

1 Department of Public Health, Aarhus University, Aarhus, Denmark, 2 Danish Cancer Society Research Center, Copenhagen, Denmark

* jiezh@ph.au.dk

## Abstract

General obesity is a recognized risk factor for various metabolically related diseases, including hypertension, dyslipidemia, and pre-diabetes. In epidemiological studies, anthropometric variables such as height and weight are often self-reported. However, misreporting of self-reported data may bias estimates of associations between anthropometry and health outcomes. Further, few validation studies have compared self-reported and measured waist circumference (WC). This study aimed to quantify the agreement between self-reported and measured height, weight, body mass index (BMI), WC, and waist-to-height ratio (WHtR), and to investigate associations of these anthropometric measures with cardiometabolic biomarkers. A total of 39,514 participants aged above 18 years were included into the Diet, Cancer, and Health-Next Generation Cohort in 2015–19. Self-reported and measured anthropometric variables, blood pressure, and cardiometabolic biomarkers (HbA1c, lipid profiles, C-reactive protein and creatinine) were collected by standard procedures. Pearson correlations ($r$) and Lin's concordance correlations were applied to evaluate misreporting. Misreporting by age, sex and smoking status was investigated in linear regression models. Multivariable regression models and Receiver Operating Characteristic analyses assessed associations of self-reported and measured anthropometry with cardiometabolic biomarkers. Self-reported height was overreported by 1.07 cm, and weight was underreported by 0.32 kg on average. Self-reported BMI and WC were 0.42 kg/m$^2$ and 0.2 cm lower than measured, respectively. Self-reported and measured height, weight, BMI, WC and WtHR were strongly correlated ($r$ = 0.98, 0.99, 0.98, 0.88, 0.86, respectively). Age, sex, smoking, and BMI contributed to misreporting of all anthropometric measures. Associations between self-reported or measured anthropometric measures and cardiometabolic biomarkers were similar in direction and strength. Concordance between self-reported and measured anthropometric measures, including WC, was very high. Self-reported anthropometric measures were reliable when estimating associations with cardiometabolic biomarkers.

**Data Availability Statement:** There are legal restrictions on sharing data that contain potentially identifying or sensitive person information, regulated by The Danish Data Protection Agency

(https://www.datatilsynet.dk). Data used in the current study will be made available upon request after application to the Diet, Cancer and Health Executive Committee. The application form can be obtained from dchdata@cancer.dk.

**Funding:** The Diet, Cancer and Health –Next generations cohort' was established with funding from the Danish Cancer Society, 'Knæk Cancer 2012' and 'Den A.P Møllerske støttefond (grant no 10619)'. This study was supported by an Aarhus University Research Foundation Starting Grant to C.C.D. (AUFF-F-2016-FLS-8-15). The funders had no role in study design, data collection and analysis, decision to publish, or preparation of the manuscript.

**Competing interests:** The authors have declared that no competing interests exist.

**Abbreviations:** **AUC**, area under the curve; **BMI**, Body mass index; **CPR**, civil registration system; **CRP**, C-reactive Protein; **DBP**, diastolic blood pressure; **DCH**, Danish Diet Cancer; **DCH-NG**, Danish Diet Cancer and Health-Next Generations; **HbA1c**, hemoglobin A1c; **HDL**, high-density lipoprotein; **IQR**, interquartile range; **LDL**, low-density lipoprotein; **LSQ**, lifestyle questionnaire; **ROC**, Receiver Operating Characteristic; **SBP**, Systolic blood pressure; **SD**, standard deviation; **TG**, triglycerides; **WC**, waist circumference; **WHtR**, waist-to-height ratio.

# 1. Introduction

In epidemiological studies, anthropometric variables are usually self-reported, especially in large study populations, as it is cost-effective and less burdensome compared to clinical measurements [1–3]. However, data may be subject to systematic errors because of observer bias or recall bias [4]. These reporting errors may be related to age, sex, race, education, and level of body mass index (BMI) [1, 4]. Several observational studies have assessed the concordance between self-reported and measured height and weight indices, and found a tendency towards overestimated height and underestimated weight, which led to an underestimate of BMI [3, 5]. The measurement error in self-reported BMI and other anthropometrics might bias estimates of associations with health outcomes, such as cardiovascular disease (CVD), type 2 diabetes (T2D), various types of cancer, and mortality [6–11]. However, only few cohorts have recorded both measured and self-reported anthropometrics at recruitment, and very little is known about how misreporting influences associations in different subgroups, for example whether there are discrepancies between younger and older participants. Furthermore, misreporting of central adiposity, as measured by waist circumference (WC) or waist-to-height ratio (WHtR), has generally not been investigated in validation studies, thus reliable comparison of self-reported and measured central adiposity-health outcome associations is lacking.

In this study, we examined the strength of association between measures and indices of general obesity, abdominal obesity, and metabolic profiles using data from the Danish Diet Cancer and Health-Next Generations (DCH-NG) cohort with adult participants. The objectives of this study were to investigate: (1) the extent of misreporting in self-reported height, weight, BMI, WC, and WHtR; (2) factors associated with misreporting; (3) comparisons of general obesity and abdominal obesity classification by self-reported and measured BMI and WC; (4) comparisons of associations of height, weight, BMI, WC, and WHtR with cardiometabolic biomarkers for self-reported and measured anthropometric measures.

# 2. Materials and methods

## 2.1 Study design and participants

The DCH-NG was established as an extension of the Danish Diet, Cancer and Health (DCH) cohort. The DCH and DCH-NG study design and methods have been described elsewhere [12, 13]. Briefly, the DCH cohort included 57,053 men and women who were born in Denmark, aged 50–64 years, had no cancer diagnosis registered in the Danish Cancer Registry and lived in the area of Copenhagen or Aarhus at the time of recruitment in 1993–1997 [13]. In 2015–2019, children of DCH cohort members (G1), their spouses (G1P) and grandchildren (G2) were invited to participate in the DCH-NG study, which aimed to investigate associations between genes, diet, and lifestyle across generations. The inclusion criteria included: individuals were listed in the Danish Civil Registration System (registers having a valid address in Denmark) [14], alive at the date of recruitment, and were above 18 years of age at the time of recruitment (born before January 1st, 2000). If a woman was pregnant at the planned time of recruitment, she was invited to participate after giving birth. Grandchildren (G2) were only invited if their parent (G1) was alive and eligible for invitation. 255,608 descendants were identified and 197,639 fulfilled the inclusion criteria. 13,875 descendants were excluded due to their status in the CPR (civil registration system, e.g. hidden address or inactive status in CPR). A total of 183,764 individuals were invited by letter and 44,869 agreed to participate in the DCH-NG (Fig 1).

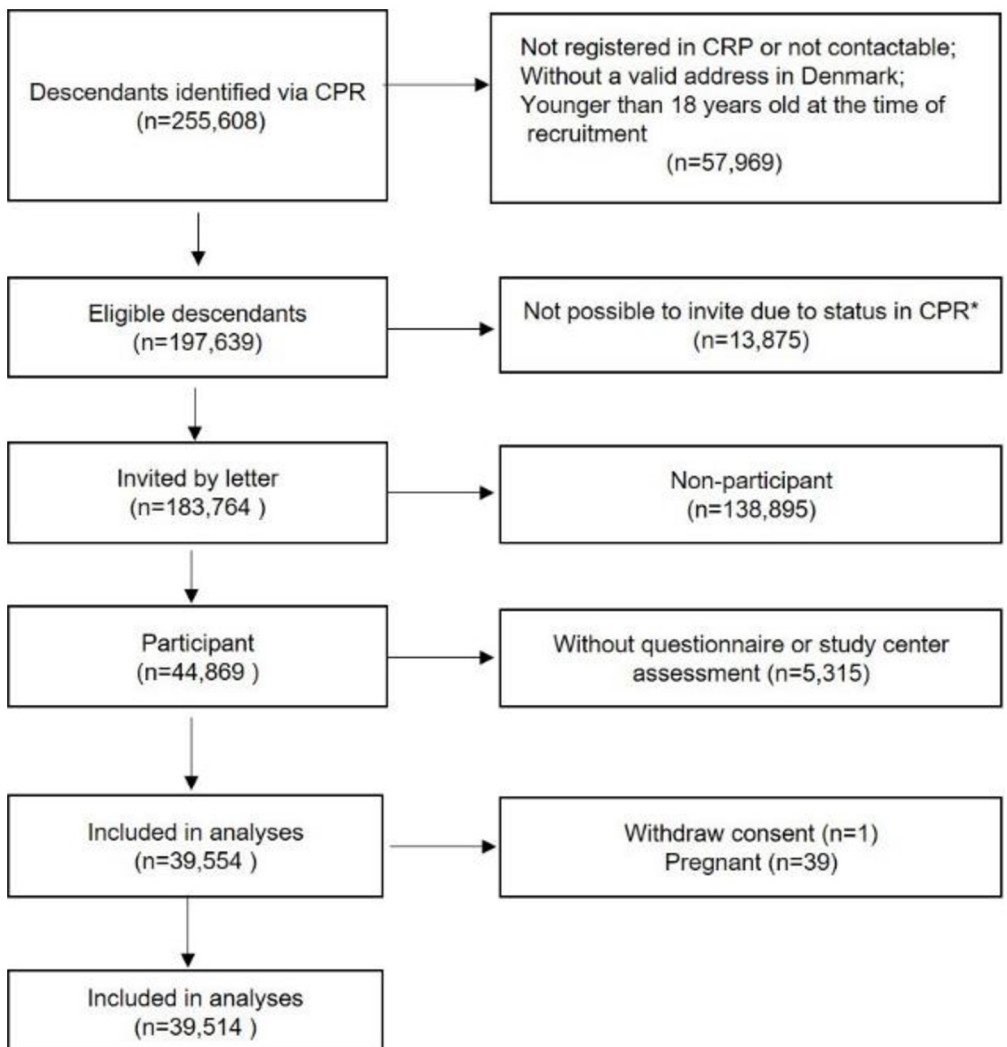

**Fig 1. Flowchart illustrating the participant route into the study.** *Status includes inactive and invalid vital status, hidden address and invalid address. CPR: civil registration system.

## 2.2 Anthropometric measurements and biomarkers

**2.2.1 Measured body weight, height, and WC.** Participants were invited to complete a physical examination in a study center in Copenhagen or Aarhus. Height, weight, and WC were measured by trained health researchers. Anthropometric measurements were assessed with participants wearing underwear and being barefoot. Height was measured to the nearest 0.1 centimeter (cm) using a wireless stadiometer (Seca 264, seca gmbh & co. kg, Hamburg, Germany). Weight was measured to the nearest 0.01 kg using a body composition analyser (Seca 515/514, seca gmbh & co. kg, Hamburg, Germany). WC was measured midway between the lower rib and the iliac crest and recorded to the nearest 0.1 cm. WC was obtained in duplicate; a third measurement was taken if the first two measures exceeded pre-specified differences (1.0 cm). The mean of the two closest measures was used for analyses. BMI was calculated using the standard formula weight (kg) divided by height squared ($m^2$), then classified into 4 categories: underweight ($<18.5$ kg/$m^2$), normal weight (18.5–24.9 kg/$m^2$),

overweight (25–29.9 kg/m$^2$), and obese ($> = 30$ kg/m$^2$) according to WHO recommendation [15]. WHtR was calculated by the formula WC in cm divided by standing height in cm. Abdominal obesity was defined by the National Institutes of Health cutoff points [16], as a WC of 102 cm or greater for men and 88 cm or greater for women. Discrepancies between self-reported and measured height of over 10 cm and between self-reported and measured weight of over 5 kg were checked for data-entry errors.

**2.2.2 Blood pressure and metabolic biomarkers.** Blood pressure was measured in a seated position after the participant had rested for at least 5 min. After measurement of the circumference of the mid-upper arm, a cuff of suitable size was applied to the participant's upper arm, which was supported by the table at heart level. Systolic blood pressure (SBP) and diastolic blood pressure (DBP) were taken three times using the Omron M-10 IT or Omron HB-1300 (OMRON Healthcare, Inc., Kyoto, Japan), and the set with the lowest values of SBP and DBP was used.

Participants were non-fasting, but were asked not to eat a fatty meal, consume alcohol, use chewing gum, brush teeth or similar within two hours before their visit to the study center. Whole blood and lithium-heparin blood samples were taken for upfront analyses performed after the visit in the study center. The following biomarkers were assessed: hemoglobin A1c (HbA1c, mmol/mol), total cholesterol (mmol/L), triglycerides (TG, mmol/L), HDL (mmol/L), low-density lipoprotein (LDL, mmol/L), C-reactive Protein (CRP, mg/L) and creatinine (μmol/L).

**2.2.3 Definition of hypertension, dyslipidemia, and pre-diabetes.** Hypertension was defined as having a blood pressure of >140/90 mmHg according to the WHO guidelines [17]. Dyslipidemia was defined as having TG >2.0 mmol/l or HDL <1.0 mmol/l based upon the recommendations by the National Heart Foundation [18] and the Australian Diabetes Society [19]. Pre-diabetes and diabetes was defined as HbA1c of >42 mmol/mol (>6.0%) [20].

**2.2.4 Lifestyle questionnaire.** Self-reported anthropometric measures were obtained from a web-based lifestyle questionnaire (LSQ), which was provided to the participants shortly after recruitment. Respondents were asked: 'How much do you weigh? Enter the nearest whole kg', 'How tall are you in cm', and 'What is your waist measurement in cm?' Respondents were provided with guidance similar to that used in the study center on where to place a tape measure for WC measurement. Smoking status was categorized as current, former, and never smoker.

## 2.3 Statistical analysis

Continuous measures are presented as mean ± standard deviation (SD) if normally distributed and median/interquartile range (IQR) if no normal distribution could be assumed. Categorical variables are shown as percentages.

Misreporting of anthropometric measures was calculated by subtracting measured from self-reported values. Bland-Altman analysis was used to compare the degree of agreement between self-reported and measured measurements, which gave the mean and 95% prediction interval of the difference between objective measurements and self-reported values [21]. Pearson's correlation coefficients (r) between self-reported and measured anthropometrics were calculated. In addition, concordance was evaluated using Lin's concordance coefficient (ρc), which is a reliability measure indicating the agreement between two measures of the same variable. To assess the agreement between measured and self-reported prevalence of general obesity and abdominal obesity, Gwet's agreement coefficient was applied. Coefficients greater than 0.81 were considered almost perfect [22]. Multiple regression analyses were used to evaluate which factors were associated with the misreporting between measured and self-reported anthropometrics (weight, height, BMI, WC, WHtR). The following variables were included in

the regression models as independent variables: sex, age groups, measured BMI categories, and smoking status.

Correlation and multivariable regression analyses were conducted to examine the cross-sectional relationships of cardiometabolic biomarkers with self-reported and measured anthropometric indices (BMI, WC, WHtR). Outcome variables with skewed distributions (TG, HbA1c, CRP, and creatinine) were log-transformed for the analyses. Model 1 was unadjusted and model 2 was adjusted for sex, age (continuous) and smoking status. Analyses were repeated using standardized coefficients. We further stratified all the analyses by different age groups to explore potential effect modification.

Receiver Operating Characteristic (ROC) analyses were conducted to assess the diagnostic accuracy of disease conditions (hypertension, dyslipidemia, and pre-diabetes) by different anthropometric measures or indices. An area under the curve (AUC) of 1 is considered to have perfect discriminatory power, and an AUC of 0.5 indicates the predictive performance is no better than chance. AUCs of 0.6–0.7 were considered poor and 0.7–0.8 fair. Optimal cut-offs on the ROC curves were chosen to maximize sensitivity and specificity of the variables with each above 50%. Sensitivity in this report was defined as the proportion of people which were categorized as positive with the disease condition, while specificity was the proportion of people which were categorized as negative without the disease condition.

**2.3.1 Sensitivity analyses.** In sensitivity analyses, we further assessed whether completing the LSQ before going to the study center or after made a difference to the results. The statistical software packages Stata/IC 16.0 (StataCorp) [23] and R (v4.0.2) [24] were used to conduct the analyses. All P-values were two-sided, and a $P < 0.05$ was considered statistically significant.

## 2.4 Ethics approval

All the procedures were performed in accordance with the ethical standards laid down in the 1964 Declaration of Helsinki and its later amendments or comparable ethical standards.

The establishment of the DCH-NG cohort was approved by the Committee on Health Research Ethics for the Capital Region of Denmark (journal number H-15001257) and by the Danish Data Protection Agency ((journal number 2013–41–2043/2014–231–0094).

The participants provided their written informed consent to participate in the DCH-NG cohort at enrolment, according to Danish Act on Research Ethics Review of Health Research Projects. The present study was approved by the Diet, Cancer and Health Scientific Management Group and no further Ethical approval or consent was needed.

# 3. Results

## 3.1 Characteristics of the study subjects

39,514 participants were included in the analyses (Fig 1). Descriptive statistics for demographic characteristics are presented in Table 1. The median age of the cohort population was 48 years (IQR: 32–54 years). 42% of the participants were men (n = 16,560). More than half of the participants reported never smoking, 32.5% of the participants were classified as being overweight and 11.9% as obese based on BMI derived from measured heights and weights. Compared with women, a larger proportion of men were overweight or obese (overweight: 42.1% male vs 25.5% female; obese: 12.7% male vs 11.3% female).

## 3.2 Factors associated with misreporting of anthropometrics

Table 2 presents the summary statistics for self-reported and measured anthropometrics. Self-reported height was 1.07 cm (SD = 1.71 cm) higher than measured height, while weight was

**Table 1. Description of demographic information of DCH-NG cohort participants.**

| | Male | | Female | | Total | |
|---|---|---|---|---|---|---|
| | Median | IQR | Median | IQR | Median | IQR |
| Age (years) | 49 | 36–55 | 47 | 30–53 | 48 | 32–54 |
| | Col % | n | Col % | n | Col % | n |
| | 42 | 16,560 | 58 | 22,954 | 100 | 39,514 |
| **Age groups** | | | | | | |
| <=25 | 11.7 | 1,930 | 15.6 | 3,574 | 13.9 | 5,504 |
| 26–35 | 12.6 | 2,083 | 13.9 | 3,190 | 13.3 | 5,273 |
| 36–45 | 12.9 | 2,138 | 13.3 | 3,049 | 13.1 | 5,187 |
| 46–55 | 38.1 | 6,317 | 39.7 | 9,124 | 39.1 | 15,441 |
| 56–65 | 23.0 | 3,814 | 17.2 | 3,956 | 19.7 | 7,770 |
| >65 | 1.7 | 278 | 0.3 | 61 | 0.9 | 339 |
| **Generation** | | | | | | |
| G1 | 53.7 | 8,898 | 53.2 | 12,221 | 53.4 | 21,119 |
| G1P | 22.1 | 3,661 | 17.3 | 3,978 | 19.3 | 7,639 |
| G2 | 24.2 | 4,001 | 29.4 | 6,755 | 27.2 | 10,756 |
| **Study center** | | | | | | |
| AAH | 28.6 | 4,741 | 28.2 | 6,481 | 28.4 | 11,222 |
| CPH | 71.4 | 11,819 | 71.8 | 16,473 | 71.6 | 28,292 |
| **Smoking status** | | | | | | |
| Current | 19.1 | 3,161 | 17.6 | 4,048 | 18.2 | 7,209 |
| Former | 26.6 | 4,405 | 29.6 | 6,796 | 28.3 | 11,201 |
| Never | 54.3 | 8,994 | 52.8 | 12,110 | 53.4 | 21,104 |
| **BMI Classification*** | | | | | | |
| Underweight | 0.7 | 112 | 2.4 | 553 | 1.7 | 665 |
| Normal | 44.5 | 7,364 | 60.7 | 13,930 | 53.9 | 21,294 |
| Overweight | 42.1 | 6,973 | 25.5 | 5,854 | 32.5 | 12,827 |
| Obese | 12.7 | 2,095 | 11.3 | 2,603 | 11.9 | 4,698 |

AAH, Aarhus; CPH, Copenhagen; DCH-NG, Diet, Cancer and Health-Next Generations; IQR, interquartile range; BMI, body mass index;

Col %, column percentage

*BMI was calculated as weight (kg) divided by height squared ($m^2$) and subsequently categorized into 4 groups according to World Health Organization criteria:

Underweight(<18.5 kg/ $m^2$); Normal(18.5 to <24.9 kg/ $m^2$); Overweight (25 to <29.9 kg/ $m^2$); Obese(> = 30 kg/$m^2$)

0.32 kg (SD = 2.15 kg) lower than measured. As a result, BMI calculated from self-reported values was 0.42 kg/$m^2$ (SD = 0.89 kg/$m^2$) lower than measured BMI. More than 77% of the deviations of self-reported BMI from measured BMI did not exceed values within 1 unit (data not shown). 33% of the participants did not report their WC, but answered 'do not know' or 'do not have a tape measure'. Participants who were male, of younger age, higher BMI and currently smoking were more likely to be missing self-reported WC. For those with available data, self-reported WC was underestimated by 0.20 cm (SD = 6.04 cm). The variances of WC were larger for both self-reported and measured values, compared with other anthropometric measures. Men tended to overreport height (1.21 cm for men and 0.97 cm for women), while women tended to underreport weight (0.11 kg for men and 0.47 kg for women). Compared to women, men had relatively lower misreporting of WC (0.12 cm for men and 0.25 cm for women). However, there was no strong evidence that WHtR was different when calculated using self-reported and measured data (Table 2). Over-reporting of height and under-reporting of weight were present across all age subgroups, but the extent of misreporting was greater

**Table 2. Description of self-reported and measured anthropometric variables in the DCH-NG cohort.**

| | | Measured values | | | Self-reported values | | | Mean difference* | | |
|---|---|---|---|---|---|---|---|---|---|---|
| | | **Mean** | **SD** | **n** | **Mean** | **SD** | **n** | **Mean** | **SD** | **n** |
| Total | Height (cm) | 173.36 | 9.17 | 39,465 | 174.45 | 9.25 | 39,403 | 1.07 | 1.71 | 39,366 |
| | Weight (kg) | 75.77 | 15.31 | 39,465 | 75.52 | 15.17 | 37,735 | -0.32 | 2.15 | 37,701 |
| | BMI (kg/m$^2$) | 25.12 | 4.24 | 39,464 | 24.68 | 4.03 | 37,662 | -0.42 | 0.89 | 37,627 |
| | WC (cm) | 87.51 | 12.73 | 39,350 | 87.29 | 12.53 | 26,159 | -0.20 | 6.04 | 26,054 |
| | WHtR | 0.50 | 0.07 | 39,329 | 0.50 | 0.07 | 26,106 | -0.00 | 0.04 | 25,997 |
| Male | Height (cm) | 181.12 | 6.68 | 16,537 | 182.34 | 6.72 | 16,525 | 1.21 | 1.81 | 16,506 |
| | Weight (kg) | 84.96 | 13.37 | 16,537 | 84.85 | 13.00 | 16,148 | -0.11 | 2.31 | 16,131 |
| | BMI (kg/m$^2$) | 25.89 | 3.78 | 16,537 | 25.50 | 3.55 | 16,127 | -0.38 | 0.89 | 16,110 |
| | WC (cm) | 93.38 | 11.75 | 16,480 | 93.70 | 11.12 | 10,521 | -0.12 | 6.56 | 10,467 |
| | WHtR | 0.52 | 0.07 | 16,472 | 0.51 | 0.06 | 10,509 | -0.00 | 0.04 | 10,454 |
| Female | Height (cm) | 167.75 | 6.15 | 22,928 | 168.75 | 6.11 | 22,878 | 0.97 | 1.62 | 22,860 |
| | Weight (kg) | 69.14 | 13.03 | 22,928 | 68.54 | 12.73 | 21,587 | -0.47 | 2.01 | 21,570 |
| | BMI (kg/m$^2$) | 24.57 | 4.47 | 22,927 | 24.07 | 4.24 | 21,535 | -0.45 | 0.88 | 21,517 |
| | WC (cm) | 83.29 | 11.69 | 22,870 | 82.98 | 11.54 | 15,638 | -0.25 | 5.67 | 15,587 |
| | WHtR | 0.50 | 0.07 | 22,857 | 0.49 | 0.07 | 15,597 | -0.00 | 0.03 | 15,543 |

SD, standard deviation; BMI, body mass index; WC, waist circumference; WHtR, waist-to-height ratio; n, number; DCH-NG, Danish Diet, Cancer, and Health -Next Generations

*Difference was calculated by subtracting measured from self-reported values

among older participants, especially for those above 55 years. Participants who were overweight or obese were more likely to misreport height, weight, and WC compared to participants of measured normal BMI. Underreporting of weight and BMI were more evident in former smokers, while WC variance was larger in current smokers (S1 Table). Multiple linear regression analyses indicated that age, sex, smoking and overweight/obese status all contributed to misreporting of all anthropometric indices (S2 Table).

## 3.3 Reliability of self-reported anthropometric variables

Comparably high correlations were found among all self-reported and measured anthropometric variables. The correlation coefficients for height, weight, BMI, WC, and WHtR were 0.98, 0.99, 0.98, 0.88, 0.86, respectively. Similarly, Lin's concordance correlations were 0.98, 0.99. 0.97, 0.88, 0.86. Bland-Altman analysis indicated no significant differences among self-reported and measured weight, height, BMI, and WC (Table 3).

BMI categories were calculated from both self-reported and measured values. The prevalence of overweight and obesity were 32.5% and 11.7% from measured BMI, but only 29.7% and 9.8% from self-reported BMI, indicating underestimation of self-reported BMI (S3 Table). Using self-reported data, 80.4% and 79.7% were correctly classified as overweight and obese (S4 Table). Agreement in classification by self-reported and measured BMI and WC, assessed by Gwet's agreement coefficients, is shown in S5 Table. When BMI was categorized into 4 groups according to WHO's criteria, Gwet's agreement coefficient between self-reported and measured BMI categories was 0.86 (95% CI: 0.86, 0.87); the agreement was 0.97 (95% CI: 0.96, 0.97) when BMI was categorized into 2 groups (normal vs overweight and obese). For WC, the agreement between self-reported and measured values was 0.84 (95% CI: 0.83, 0.85).

**Table 3. Correlation and concordance between self-reported and measured anthropometrics.**

| | obs n | r | ρc | Bland-Altman | | |
| --- | --- | --- | --- | --- | --- | --- |
| | | | | Average | 95% CI | |
| Height (cm) | 39,366 | 0.98 | 0.98 | 1.07 | -2.28 | 4.42 |
| Weight (kg) | 37,701 | 0.99 | 0.99 | -0.32 | -4.53 | 3.90 |
| BMI (kg/m2) | 37,627 | 0.98 | 0.97 | -0.42 | -2.16 | 1.32 |
| WC (cm) | 26,054 | 0.88 | 0.88 | -0.20 | -12.05 | 11.65 |
| WHtR | 25,997 | 0.86 | 0.86 | 0.00 | -0.07 | 0.07 |

BMI, body mass index; WC, waist circumference; WHtR, waist-to-height ratio; CI, confidence intervals; n, number

r, pearson correlation

ρc, Lin's concordance

### 3.4 Associations between anthropometric measures and cardiometabolic biomarkers

The associations between anthropometric measures and metabolic biomarkers are presented as standardized (S6 Table) and unstandardized (S7 Table) linear coefficients. There were positive associations between the measures and TG, LDL, HbA1c, total cholesterol concentration, and blood pressure, respectively, and an inverse relationship with HDL concentration in both unadjusted and adjusted models. Using standardized coefficients, the associations with metabolic biomarkers did not vary substantially when comparing self-reported and measured indices, although associations were stronger for measured WHtR and measured WC than for self-reported WHtR and WC (Fig 2). In stratified analyses, positive associations when examining central adiposity on total cholesterol, LDL, SBP and DBP were observed for early adulthood to

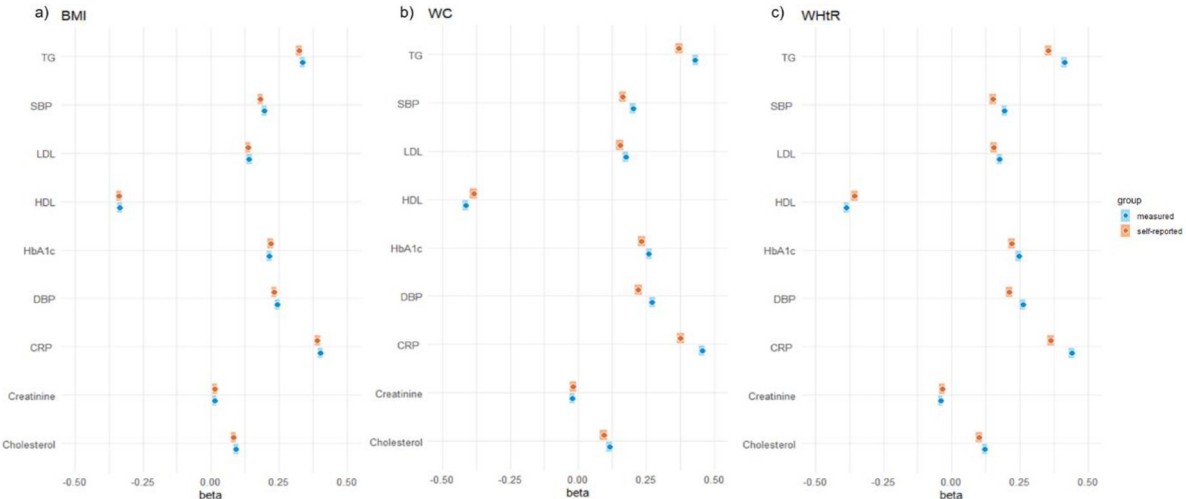

**Fig 2. Standardized coefficients of self-reported or measured anthropometrics and cardiometabolic biomarkers.** a) self-reported (orange color) or measured (blue color) BMI associated with cardiometabolic biomarkers. b) self-reported (orange color) or measured (blue color) WC associated with cardiometabolic biomarkers. c) self-reported (orange color) or measured (blue color) WHtR associated with cardiometabolic biomarkers. Models were adjusted for age, sex, and smoking. TG, triglycerides; HDL, high-density lipoprotein; LDL, low-density lipoprotein; HbA1c, hemoglobin A1c; CRP, C-reactive Protein; SBP, systolic blood pressure; DBP, diastolic blood pressure; BMI, body mass index; WC, waist circumference; WHtR, waist-to-height ratio; CVD, cardiovascular diseases.

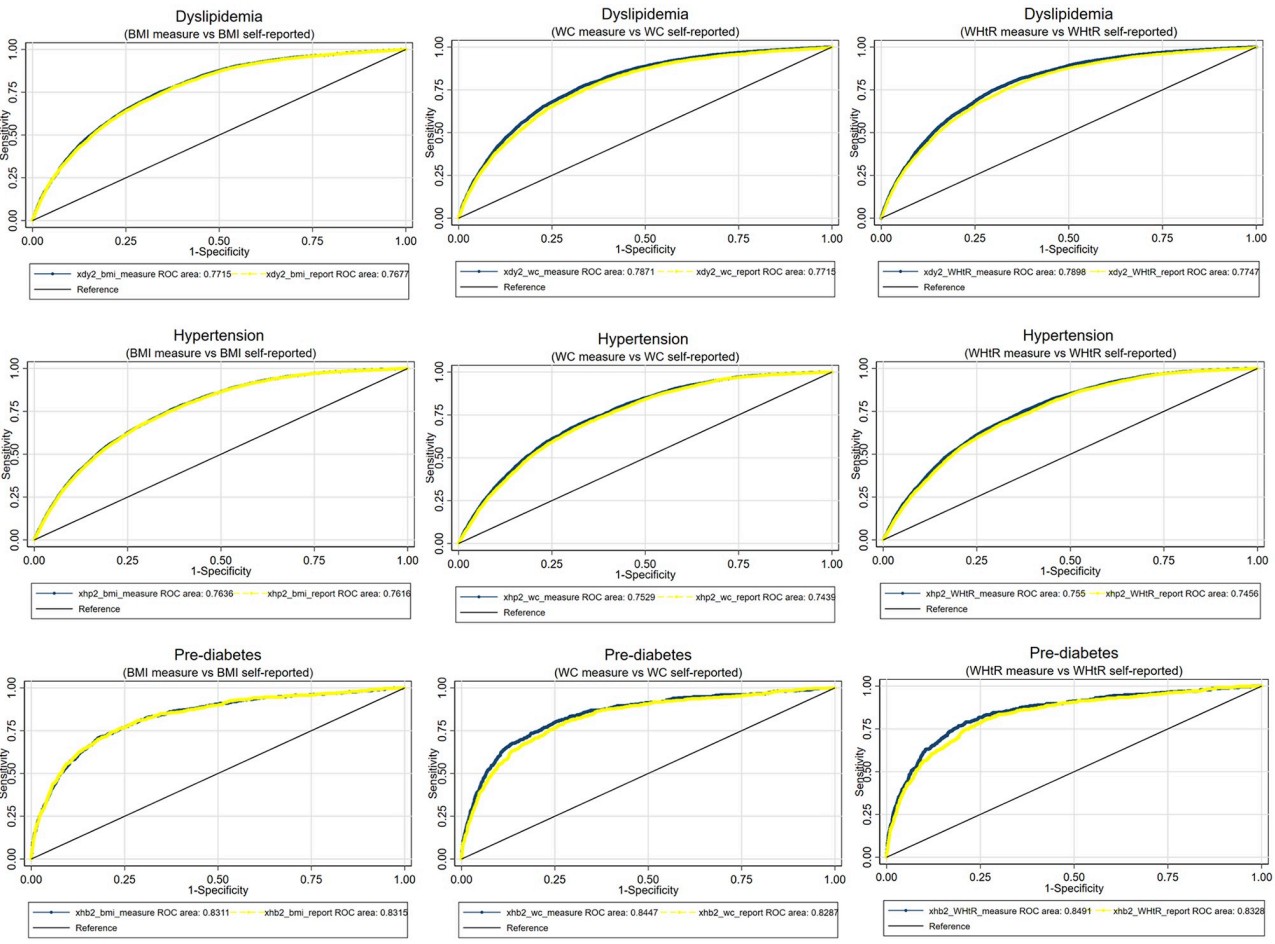

**Fig 3. ROC curves comparing measured and self-reported anthropometric indices with cardiometabolic biomarkers.** Green lines indicate measured anthropometrics, yellow lines indicate self-report anthropometrics, black lines indicate reference. Models were adjusted for age, sex, and smoking. ROC, Receiver Operating Characteristic.

middle-aged groups, whereas associations were weaker for participants above 55 years of age (S8 Table).

The prevalence of hypertension, dyslipidemia, and pre-diabetes were 23.2%, 5.5%, and 1.7%, respectively. The AUC for each anthropometric index with hypertension and dyslipidemia indicated fair predictive performance, ranging between 0.7–0.8; prediction was excellent for pre-diabetes (above 0.8; Fig 3, S9 Table).

In sensitivity analyses, the correlation between self-reported and measured height and weight was equal regardless of whether participants completed the LSQ before going to the study center or after, whereas the correlation between self-reported and measured WC was slightly higher among those completing the LSQ after going to the study center (0.93 vs 0.88) (S10 Table).

## 4. Discussion

In this large contemporary cohort, we found high agreement between self-reported and measured anthropometrics, albeit slightly lower for WC and WHtR than for other measures. Older age, male sex, weight status, and smoking contributed to misreporting. We compared the

associations between self-reported or measured anthropometric variables and cardiometabolic biomarkers, and found similar directions and magnitudes across all age groups. Our results extend previous validation studies that only included certain specific age, ethnic, or BMI groups [25–32], which may limit generalizability to general population studies.

Our findings are in agreement with previous European validation studies [33–36]. Self-reported BMI may bias associations with health outcomes, despite high correlations between self-reported and measured values [6]. A US cohort study compared associations between self-reported and measured BMI and cardiometabolic biomarkers in young adults, and found similar strengths of association [3], as have studies in US adults [5, 7]. However, other studies have reported poor performance for self-reported anthropometric indices in predicting cardiometabolic related diseases [9, 37]. These differences may be explained by study design, different age groups, and approaches in adjusting for confounders. We found that the strength of associations with cardiometabolic biomarkers did not vary substantially for measured and self-reported indices. While associations were stronger for measured indices compared with self-reported, the bias was small.

We observed greater misreporting among older age groups, as well as differences in strengths of association between both measured and self-reported anthropometrics and cardiometabolic biomarkers across age groups. These results are, however, comparable to results from the WHO MONICA Project, which found that the association between BMI and hypercholesterolemia (cholesterol ≥6.5 mmol/l) was weaker in higher age groups [38]. Similar results were reported from a large scale Chinese study [39]. A possible explanation could be increasing fat mass and lower lean mass with increasing age [40].

Our study has several strengths. With self-reported and measured WC, we provide much needed assessment of central adiposity indices [41]. We included a wide age range, which allowed investigation of age on misreporting and associations. Finally, the DCH-NG cohort is a new cohort [12], and thus reflects contemporary trends in self-assessment of weight, height and WC.

The study also has limitations. Firstly, similar to the DCH cohort, participants in the DCH-NG cohort had a higher socioeconomic status relative to the nonparticipants [13]. Therefore, it is important to take into account the generalizability and representativeness of the current sample in regard to other study populations and the general population when interpreting the results. Secondly, there was a higher proportion of missing data in self-reported WC than for other measures. Our sensitivity analyses suggested that younger people, those with greater BMI and who smoked tended not to report WC. However, the missingness was not related to measured WC values, and there was little missing data among those participants who completed the LSQ after having visited the study facility. It is likely that participants did not report WC values simply because they did not know their WC. Improving general knowledge about WC and related health outcomes may be useful from a public health perspective [42]. Lastly, some participants had a longer gap between filling in the questionnaires and the clinical appointments than others, which might influence the consistency of the measures. However, the percentage of participants for whom this was the case was small and thus unlikely to bias the results.

## 5. Conclusion

This study found a high correlation between self-reported and measured anthropometric variables across a wide age range in the study population. The overall agreements with respect to the reporting of accurate values were reliable and acceptable. However, caution may need to be taken when generalizing the findings to other populations.

## Supporting information

**S1 Table. Misreporting on anthropometric variables with participant characteristics.**
(DOCX)

**S2 Table. Factors associated with misreporting on anthropometric variables in multivariable regression models.**
(DOCX)

**S3 Table. BMI classification based on measured and self-reported BMI.**
(DOCX)

**S4 Table. Cross-classification of measured and self-reported BMI classification.**
(DOCX)

**S5 Table. Agreement between self-reported indices compared to measured indices.**
(DOCX)

**S6 Table. Self-reported and measured anthropometric variables with CVD biomarkers\* (Standarised).**
(DOCX)

**S7 Table. Self-reported and measured anthropometric variables with CVD biomarkers\* (Unstandarised).**
(DOCX)

**S8 Table. Self-reported and measured anthropometric variables with CVD biomarkers stratified by age.**
(DOCX)

**S9 Table. ROC curve comparing measured and self-reported anthropometric indices with cardiovascular risk factors.**
(DOCX)

**S10 Table. Correlation between self-reported and measured anthropometrics stratified by visiting time.**
(DOCX)

## Acknowledgments

The authors express sincere appreciation to all the participants who took part in the study and the Danish Cancer Society and staff at the Diet, Cancer and Health study for the collection and administration of data.

## Author Contributions

**Conceptualization:** Jie Zhang, Anja Olsen, Jytte Halkjær, Kristina E. Petersen, Anne Tjønneland, Kim Overvad, Christina C. Dahm.

**Data curation:** Jie Zhang, Jytte Halkjær, Kristina E. Petersen, Anne Tjønneland, Kim Overvad, Christina C. Dahm.

**Formal analysis:** Jie Zhang, Kristina E. Petersen.

**Funding acquisition:** Anja Olsen, Jytte Halkjær, Anne Tjønneland, Kim Overvad.

**Investigation:** Jie Zhang, Kim Overvad, Christina C. Dahm.

**Methodology:** Jie Zhang, Anja Olsen, Jytte Halkjær, Kristina E. Petersen, Kim Overvad, Christina C. Dahm.

**Project administration:** Anja Olsen, Jytte Halkjær, Kristina E. Petersen, Anne Tjønneland, Kim Overvad, Christina C. Dahm.

**Resources:** Anne Tjønneland, Christina C. Dahm.

**Software:** Jie Zhang.

**Supervision:** Anja Olsen, Jytte Halkjær, Anne Tjønneland, Kim Overvad, Christina C. Dahm.

**Validation:** Jie Zhang, Jytte Halkjær, Kristina E. Petersen, Christina C. Dahm.

**Visualization:** Jie Zhang.

**Writing – original draft:** Jie Zhang.

**Writing – review & editing:** Anja Olsen, Jytte Halkjær, Kristina E. Petersen, Anne Tjønneland, Kim Overvad, Christina C. Dahm.

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
