## [Decision Letter · Decision Letter 0]

30 Mar 2023

PONE-D-22-34141Self-reported and measured anthropometric variables in association with cardiometabolic markers: a Danish cohort studyPLOS ONE

Dear Dr. Jie Zhang,

Thank you for submitting your manuscript to PLOS ONE. After careful consideration, we feel that it has merit but does not fully meet PLOS ONE’s publication criteria as it currently stands. Therefore, we invite you to submit a revised version of the manuscript that addresses the points raised during the review process.

Please submit your revised manuscript by May 14 2023 11:59PM.  If you will need more time than this to complete your revisions, please reply to this message or contact the journal office at plosone@plos.org. Please include the following items when submitting your revised manuscript:A rebuttal letter that responds to each point raised by the academic editor and reviewer(s). You should upload this letter as a separate file labeled 'Response to Reviewers'.A marked-up copy of your manuscript that highlights changes made to the original version. You should upload this as a separate file labeled 'Revised Manuscript with Track Changes'.An unmarked version of your revised paper without tracked changes. You should upload this as a separate file labeled 'Manuscript'.If applicable, we recommend that you deposit your laboratory protocols in protocols.io to enhance the reproducibility of your results. Protocols.io assigns your protocol its own identifier (DOI) so that it can be cited independently in the future. For instructions see: https://journals.plos.org/plosone/s/submission-guidelines#loc-laboratory-protocols. Additionally, PLOS ONE offers an option for publishing peer-reviewed Lab Protocol articles, which describe protocols hosted on protocols.io. Read more information on sharing protocols at https://plos.org/protocols?utm_medium=editorial-email&utm_source=authorletters&utm_campaign=protocols.

We look forward to receiving your revised manuscript.

Kind regards,

Reindolf Anokye

Academic Editor

PLOS ONE

Journal Requirements:

" ‘The Diet, Cancer and Health –Next generations cohort’ was established with funding from the Danish Cancer Society, ‘Knæk Cancer 2012’ and ‘Den A.P Møllerske støttefond (grant no 10619)’.

No financial disclosures were reported by the authors of this paper."

Reviewers' comments:

Reviewer's Responses to Questions

**Comments to the Author**

1. Is the manuscript technically sound, and do the data support the conclusions?

Reviewer #1: Yes

Reviewer #2: Yes

2. Has the statistical analysis been performed appropriately and rigorously? 

Reviewer #1: Yes

Reviewer #2: Yes

3. Have the authors made all data underlying the findings in their manuscript fully available?

Reviewer #1: Yes

Reviewer #2: No

4. Is the manuscript presented in an intelligible fashion and written in standard English?

Reviewer #1: Yes

Reviewer #2: Yes

5. Review Comments to the Author

Reviewer #1: Dear Editor

I read interestingly the manuscript entitled “Self-reported and measured anthropometric variables in association with cardiometabolic markers: a Danish cohort study”. The study seems conducted carefully.

- Please prepare a list of abbreviations at the beginning of the Meta DATA.

Abstracts

-It is not clear in the abstract that why should we find an association between anthropometric indices and CVD diseases after we calibrated them, please justify it more clearly

- Keywords: are these keywords are Mesh terms? Word that serves as a keyword, as to the meaning of that condition must be a Mesh term

Introduction

- I am not sure about this claim of authors do they have any valid refrernce for it?

“In epidemiological studies, anthropometric variables are usually self-reported, especially in large study populations, as it is cost-effective and less burdensome compared to clinical measurements”.

Line 114: the authors started to discuss about blood pressure but they did not mention about it in the abstract

Results

There is no table that summarize demographic data of included participants, it would be better that authors report a table in this regard

Discussion

- The authors should list and shortly discuss the limitation of their study, for instance their the lack of evidence about justification of this research

Reviewer #2: Although research idea is not novel and have been searched before . The authors made a great effort to re investigate the idea and correlated the results with cardiac biomarker. Regarding the sample size how did you calculate it and why didn't you mention it.

6. PLOS authors have the option to publish the peer review history of their article (what does this mean?). If published, this will include your full peer review and any attached files.

Reviewer #1: No

Reviewer #2: No

---

## [Author Response · Author response to Decision Letter 0]

20 Jun 2023

Dear Editors and Reviewers,

Thank you for the detailed comments and the insightful suggestions to improve our manuscript. We have edited the manuscript to incorporate changes that reflect the detailed suggestions you have kindly provided. To facilitate your review of our revisions, the following is a point-by-point response to the questions and comments. 

Dear Editors and Reviewers,

Thank you for the detailed comments and the insightful suggestions to improve our manuscript. We have edited the manuscript to incorporate changes that reflect the detailed suggestions you have kindly provided. To facilitate your review of our revisions, the following is a point-by-point response to the questions and comments. 

Reviewer #1: Dear Editor

I read interestingly the manuscript entitled “Self-reported and measured anthropometric variables in association with cardiometabolic markers: a Danish cohort study”. The study seems conducted carefully.

- Please prepare a list of abbreviations at the beginning of the Meta DATA.

Re: The abbreviations were provided at the end of the manuscript, please refer to line 357-375. 

Abstracts

-It is not clear in the abstract that why should we find an association between anthropometric indices and CVD diseases after we calibrated them, please justify it more clearly

Re: Thank you for the comment. We have revised the abstract to emphasize the rational to study the association between anthropometric indices and CVD outcomes more clearly. A new sentence has been added in the abstract (Line 27-30). 

Line 27-30: General obesity is a recognized risk factor for various metabolically related diseases, including hypertension, dyslipidemia, and pre-diabetes. In epidemiological studies, anthropometric variables such as height and weight are often self-reported.

- Keywords: are these keywords are Mesh terms? Word that serves as a keyword, as to the meaning of that condition must be a Mesh term

Re: Thank you for pointing this out, we have reviewed and updated the keywords to ensure they are Mesh terms. 

Introduction

- I am not sure about this claim of authors do they have any valid refrernce for it?

“In epidemiological studies, anthropometric variables are usually self-reported, especially in large study populations, as it is cost-effective and less burdensome compared to clinical measurements”.

Re: Thank you for raising the concern. After further review, we have added several studies that have supported this claim, including a systematic review [1], and some large cohort studies[2, 3]. We have now included these references in our manuscript to strengthen the validity of our claim. 

Line 114: the authors started to discuss about blood pressure but they did not mention about it in the abstract

Re: Thank you so much for pointing this out. We have revised the abstract to make it clear. 

Before After

Self-reported and measured anthropometric variables, as well as cardiometabolic biomarkers, were obtained from participants aged aboved 18 years at recruitment to the Diet, Cancer, and Health-Next Generation Cohort in 2015-19 (N=39,514). Line 35-38: A total of 39,514 participants aged above 18 years were included into the Diet, Cancer, and Health-Next Generation Cohort in 2015-19. Self-reported and measured anthropometric variables, blood pressure, and cardiometabolic biomarkers (HbA1c, lipid profiles, C-reactive protein and creatinine) were collected by standard procedures.

Results

There is no table that summarize demographic data of included participants, it would be better that authors report a table in this regard

Re: Appendix Table 1 is a description of demographic information of the participants, which was provided in supplementary information. To make it clear, we have moved this table to the main manuscript as Table 1, and the table numbers have been updated accordingly. 

Discussion

- The authors should list and shortly discuss the limitation of their study, for instance their the lack of evidence about justification of this research.

Re: We appreciate your feedback and agree that it is important to acknowledge the limitation of the study. 

In terms of the lack of evidence about the justification of our research, we would like to clarify that our study aimed to address the concern regarding misreporting in self-reported measures of anthropometry and its influence on estimates of associations between anthropometry and health outcomes. We have elaborated the justification of this study in the introduction (Line 57-72) and discussion (Line 286-296). 

We acknowledge there are limitations of the study, for instance, the generalizability and representativeness of the cohort (Line 312-325).

Reviewer #2: Although research idea is not novel and have been searched before . The authors made a great effort to re investigate the idea and correlated the results with cardiac biomarker. Regarding the sample size how did you calculate it and why didn't you mention it.

Re: Thank you for the feedback and comments. This validation study was based on the Danish Diet, Cancer, and Health-Next Generation cohort (DCH-NG), with the purpose of examining whether self-reported anthropometrics are comparable with measured anthropometrics in terms of associated with cardiometabolic outcomes. 

The DCH-NG is a population based cohort study. The sample size was calculated based on the number of cancer cases and other health outcomes expected to develop since 2011 as well as the magnitude of the relative risk to be detected, under the scenario with an estimated response rate of 30%. The sample size was determined to achieve a power of 80% and a significance level of 5%. The sample size calculation was based on the expected effect size and the variability of the target variables in the population. For details, please refer to the DCH-NG cohort description[4]. 

6. PLOS authors have the option to publish the peer review history of their article (what does this mean?). If published, this will include your full peer review and any attached files.

Do you want your identity to be public for this peer review? For information about this choice, including consent withdrawal, please see our Privacy Policy.

Reviewer #1: No

Reviewer #2: No

Reference

1. Maukonen M, Männistö S, Tolonen H. A comparison of measured versus self-reported anthropometrics for assessing obesity in adults: a literature review. Scandinavian journal of public health. 2018;46(5):565-79.

2. Hodge JM, Shah R, McCullough ML, Gapstur SM, Patel AV. Validation of self-reported height and weight in a large, nationwide cohort of US adults. PloS one. 2020;15(4):e0231229.

3. Lipsky LM, Haynie DL, Hill C, Nansel TR, Li K, Liu D, et al. Accuracy of self-reported height, weight, and BMI over time in emerging adults. American journal of preventive medicine. 2019;56(6):860-8.

4. Petersen KE, Halkjær J, Loft S, Tjønneland A, Olsen A. Cohort profile and representativeness of participants in the Diet, Cancer and Health—Next Generations cohort study. European journal of epidemiology. 2022:1-11.

---

## [Decision Letter · Decision Letter 1]

10 Jul 2023

Self-reported and measured anthropometric variables in association with cardiometabolic markers: a Danish cohort study

PONE-D-22-34141R1

Dear Dr. Jie Zhang,

We’re pleased to inform you that your manuscript has been judged scientifically suitable for publication and will be formally accepted for publication once it meets all outstanding technical requirements.

Kind regards,

Reindolf Anokye

Academic Editor

PLOS ONE

Additional Editor Comments (optional):

Reviewers' comments:

Reviewer's Responses to Questions

**Comments to the Author**

1. If the authors have adequately addressed your comments raised in a previous round of review and you feel that this manuscript is now acceptable for publication, you may indicate that here to bypass the “Comments to the Author” section, enter your conflict of interest statement in the “Confidential to Editor” section, and submit your "Accept" recommendation.

Reviewer #1: All comments have been addressed

2. Is the manuscript technically sound, and do the data support the conclusions?

Reviewer #1: (No Response)

3. Has the statistical analysis been performed appropriately and rigorously? 

Reviewer #1: (No Response)

4. Have the authors made all data underlying the findings in their manuscript fully available?

Reviewer #1: (No Response)

5. Is the manuscript presented in an intelligible fashion and written in standard English?

Reviewer #1: (No Response)

6. Review Comments to the Author

Reviewer #1: All mentioned comments have been addressed and I am satisfied with the response of the authors, thank you

7. PLOS authors have the option to publish the peer review history of their article (what does this mean?). If published, this will include your full peer review and any attached files.

Reviewer #1: No

---

## [Editor Report · Acceptance letter]

17 Jul 2023

PONE-D-22-34141R1 

Self-reported and measured anthropometric variables in association with cardiometabolic markers: a Danish cohort study 

Dear Dr. Zhang:

I'm pleased to inform you that your manuscript has been deemed suitable for publication in PLOS ONE. Congratulations! Your manuscript is now with our production department. 

Kind regards, 

on behalf of

Dr Reindolf Anokye 

Academic Editor

PLOS ONE